# Spontaneous formation of structurally diverse membrane channel architectures from a single antimicrobial peptide

Yukun Wang[1,2,3], Charles H. Chen[3,4], Dan Hu[1,5], Martin B. Ulmschneider[3,4] & Jakob P. Ulmschneider[1,2]

Many antimicrobial peptides (AMPs) selectively target and form pores in microbial membranes. However, the mechanisms of membrane targeting, pore formation and function remain elusive. Here we report an experimentally guided unbiased simulation methodology that yields the mechanism of spontaneous pore assembly for the AMP maculatin at atomic resolution. Rather than a single pore, maculatin forms an ensemble of structurally diverse temporarily functional low-oligomeric pores, which mimic integral membrane protein channels in structure. These pores continuously form and dissociate in the membrane. Membrane permeabilization is dominated by hexa-, hepta- and octamers, which conduct water, ions and small dyes. Pores form by consecutive addition of individual helices to a transmembrane helix or helix bundle, in contrast to current poration models. The diversity of the pore architectures—formed by a single sequence—may be a key feature in preventing bacterial resistance and could explain why sequence–function relationships in AMPs remain elusive.

[1] Institute of Natural Sciences, Shanghai Jiao-Tong University, 800 Dongchuan Road, Shanghai 200240, China. [2] Department of Physics and Astronomy, Shanghai Jiao-Tong University, 800 Dongchuan Road, Shanghai 200240, China. [3] Institute of NanoBioTechnology, Johns Hopkins University, 204C Schaffer Hall, 3400 North Charles Street, Baltimore, Maryland 21218-2681, USA. [4] Department of Materials Science and Engineering, Johns Hopkins University, 204C Schaffer Hall, 3400 North Charles Street, Baltimore, Maryland 21218-2681, USA. [5] Department of Mathematics, Shanghai Jiao-Tong University, 800 Dongchuan Road, Shanghai 200240, China. Correspondence and requests for materials should be addressed to M.B.U. (email: martin@ulmschneider.com) or to J.P.U. (email: jakob@sjtu.edu.cn).

Since their discovery over 100 years ago, thousands of pore-forming antimicrobial peptides (AMPs) have been identified, revealing great variations in size, secondary structure and sequence composition[1]. Despite this wealth of data no common pore-forming motif has been discovered, and the molecular basis of antimicrobial activity remains poorly understood[2,3]. One of the key difficulties has been the transience of pores formed by AMPs, which has proved challenging for experimental structure determination. In the absence of experimental structures a range of theoretical models have been proposed that try to rationalize how soluble AMPs, which carry multiple charged residues, can insert into hydrophobic membranes to form pores and what these pores might look like[2].

Here we report the molecular mechanisms of pore formation, as well as the structures, conductances and selectivities of pores formed by maculatin (GLFGVLAKVAAHVVPAIAEHF-NH₂), a 21-residue peptide isolated from the skin of the green-eyed tree frog, *Litoria genimaculata*. Maculatin is a typical member of the large family of pore-forming amphiphilic AMPs, with broad-spectrum antibacterial activity at low micromolar concentrations. Previous studies have shown that maculatin forms long-lived pores, which cause all-or-none leakage comparable to other AMPs[4–7].

## Results

**Experimentally guided system set-up**. On contact with phosphatidylcholine liposomes, maculatin peptides, which are unstructured in solution, fold into continuous α-helices that form pores in the liposome membrane. First, we used circular dichrosim (CD) spectroscopy to establish experimentally that the P15A mutation, which is active against Gram-positive bacteria[4,8,9], stabilizes membrane-bound maculatin against thermal denaturation without affecting pore size or pore-forming capactiy for a range of different lipids and

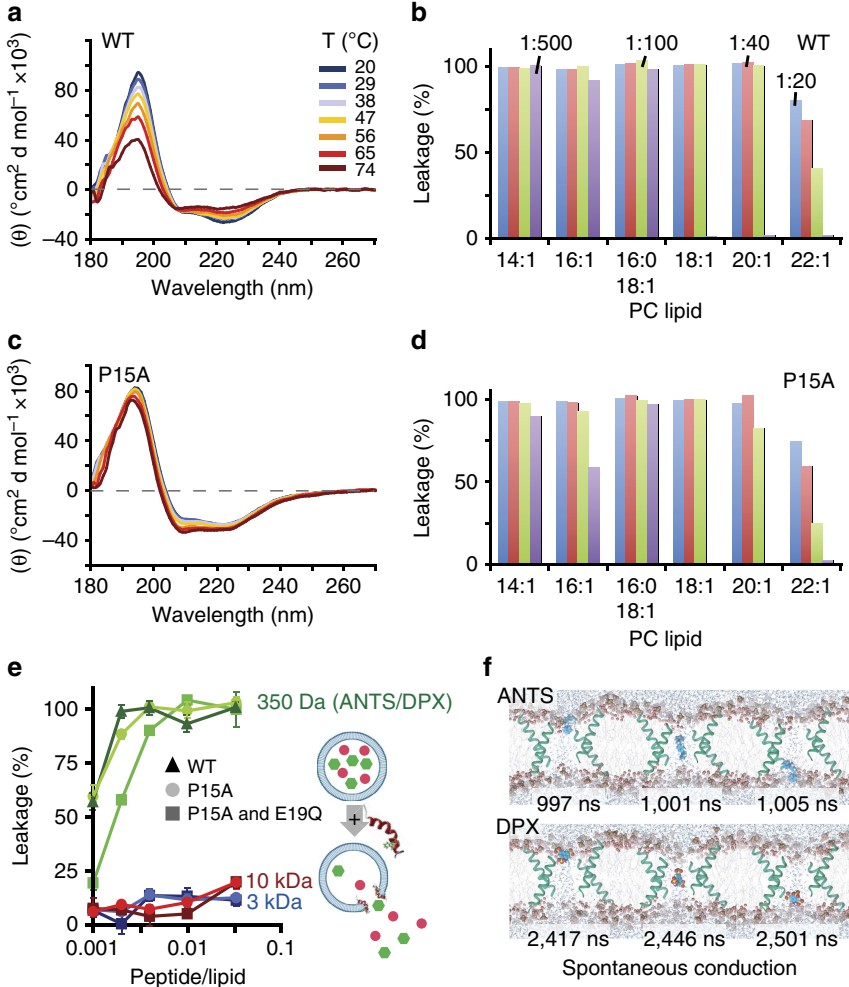

**Figure 1 | Thermally stabilized maculatin.** (**a**) CD spectra show that wild-type (WT) maculatin starts to denature at elevated temperatures in the presence of 100 nm POPC LUVs (peptide-to-lipid ratio = 1/100). (**b**) WT maculatin-induced LUV leakage is reduced for bilayers containing phosphatidylcholine (PC) lipids with longer hydrophobic tails (14:1 Δ9-cis; 16:1 Δ9-cis; 18:1 Δ9-cis; 20:1 Δ11-cis; 22:1 Δ13-cis). (**c**) The single-mutation P15A stabilizes maculatin against thermal denaturation, with no detectable loss of helicity even at 74 °C. The temperatures shown were the temperature measured for the cuvette, with 74 °C corresponding to a cell-holder temperature of 95 °C, the highest setting possible. (**d**) P15A induces similar but less liposomal leakage as WT maculatin, with similar lipid tail length dependence. (**e**) The pore-sizing assay measures the leakage of dyes of increasing size (400–10,000 Da) from 0.5 mM POPC LUVs (diameter = 100 nm) after addition of 0.5 μM peptide (that is, P/L = 1/1,000) using fluorescence spectroscopy. Hundred per cent leakage was determined using 10 vol.% Triton X-100. Pore size and leakage efficiency of the P15A single and P15A-E19Q double mutants are similar to that of WT. (**f**) Octameric pores formed by P15A-E19Q during assembly simulations were found to spontaneously conduct both ANTS and DPX dyes using unbiased conductance simulations. Error bars are s.e.m.

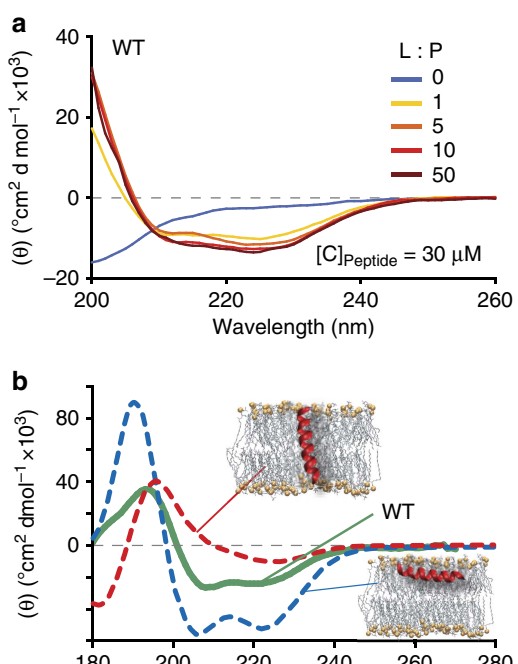

**Figure 2 | Experimental partitioning free energy and average membrane orientation of maculatin.** (**a**) CD spectra of the titration of 30 μM maculatin wild type (WT) and mutants with 100 nm POPC LUVs, show that membrane binding is strong $\Delta G_{binding} = -4.6 \pm 0.8$ kcal mol$^{-1}$. Consistent with the experimental binding the simulations show that at the molecular-level membrane binding is concomitant with interfacial folding of the peptide into a surface-absorbed alpha helix. (**b**) Oriented CD spectra of WT maculatin in POPC bilayer stacks at 100% relative humidity is $50 \pm 15\%$ TM-inserted (peptide-to-lipid ratio = 1/30). The dashed lines show theoretical helical spectra for peptides aligned perfectly parallel (red) and perfectly perpendicular (blue) to the beam, corresponding to TM and surface-bound peptides, respectively.

peptide-to-lipid ratios (Fig. 1). This thermally stabilized mutant allowed us to elevate the temperatures of unbiased long-timescale folding-partitioning molecular dynamics simulations to 90–150 °C, increasing sampling by 2–3 orders of magnitude, without significantly affecting the thermodynamics of the system[10–14]. Next, we experimentally determined the leakage of fluorescent dyes of increasing molecular weight from liposomes treated with maculatin (Fig. 1). This revealed that only the smallest dyes encapsulated (8-aminonaphthalene-1,3,6-trisulphonic acid/p-Xylene-bis-N-pyridinium bromide; ANTS/DPX), with molecular weights of ~400 Da, are able to leak out of the liposome through pores formed by maculatin peptides, allowing us to estimate the approximate number of peptides forming a pore, by assuming simple channel geometries. Using this information, we built atomic detail systems of up to 16 maculatin peptides placed in phosphatidylcholine lipid bilayers of different acyl chain lengths (Supplementary Table 1).

**Maculatin membrane insertion and oligomerization.** All maculatin peptides initially reside in their stable surface-adsorbed state[15], consistent with the $-4.6 \pm 0.8$ kcal mol$^{-1}$ partitioning free energy determined for wild-type maculatin via CD titration (Fig. 2). Figure 3 shows that individual peptides subsequently insert transmembrane (TM) via a range of different mechanisms. This finding was unexpected, since TM insertion of AMPs like maculatin requires temporary burial and membrane translocation of charged side chains, which have theoretical membrane

translocation barriers of ~15–20 kcal mol$^{-1}$ (ref. 16). In the dominant insertion mechanism this barrier is overcome by cooperative insertion, involving two peptides in a head-to-tail arrangement in combination with a water defect. TM-inserted maculatin can rapidly catalyse additional TM insertions through membrane defects induced by its charged and polar side chains. In contrast to some proposed models, AMP insertion via large surface aggregates was not observed[17,18]. At equilibrium the peptides, which were initially placed on one interface in simulations DM1–DM4, are symmetrically distributed along the bilayer normal, continually changing between marginally stable TM oligomeric assemblies and surface-bound states on both interfaces (Fig. 3). Subsequent pore-forming simulations were therefore started with an equal number of peptide on each interface.

**Spontaneous formation of structurally diverse pores.** Pore formation indicates that a significant number of peptides are TM-inserted. Overall, the surface-bound states slightly dominate the equilibrium with $53 \pm 7\%$ occupation (averaged over all simulations $\Delta G_{S \to TM} = 0.1$ kcal mol$^{-1}$), in agreement with the $50 \pm 15\%$ interfacial population determined from oriented CD measurements in aligned palmitoyloleoyl-phosphocholine (POPC) bilayer stacks (Fig. 2). Figure 4 shows that 10–20 μs timescales capture numerous spontaneous pore-formation events. Remarkably, rather than a single, well-defined and stable pore structure, as would be expected for a channel protein, maculatin forms an ensemble of conformationally diverse channel-like pores, which continually assemble and disband in the membrane (Fig. 4). The formation of an equilibrium ensemble of structurally diverse channels by a single AMP has not been previously reported in the literature, but is consistent with experimental leakage data.

To assess the architectural diversity of the channel-like pores formed, we developed software to identify and cluster structurally similar oligomeric TM assemblies (see Methods). This revealed hundreds of different channel architectures for maculatin, of which usually <10 are significantly populated. Only small pores with 3–8 peptides were observed. Larger aggregates were found only rarely and consisted of smaller channels in lateral contact. The most populated cluster for each oligomer are shown in Figs 4 and 5. Interestingly, very similar oligomers were found for all simulations, with a strong preference for antiparallel peptide arrangements (90%). The recurring structural motif is an antiparallel dimer offset by 8–10 residues, with tight inter-helical packing of the C-terminal moieties of the peptides (Supplementary Fig. 1). Larger oligomers are symmetric combinations of this basic dimer motif.

Remarkably, single (P15A; E19Q) and double mutants (P15A, E19Q), which show small or little difference in experimental leakage and pore size (Fig. 1), revealed different ensemble weightings (Fig. 4), as did changes in membrane acyl chain length (Fig. 5). This shows that while maculatin forms proper channels in the membrane, rather than disordered pores or detergent-like holes, the channel equilibrium is influenced by the environment or minor mutations. Indeed, the heterogeneous nature of the channel-like structures precludes their classification into the generally assumed barrel stave or toridal models suggested for AMP pore structures[2,19,20].

Unbiased dye-conductance simulations revealed that the octameric pores efficiently conduct ANTS/DPX fluorophores, which were the largest dyes observed to leak from vesicles treated by maculatin (Fig. 1). This confirms that these are indeed the largest pores that maculatin can form. The diversity of pores observed, and their variation on minor perturbation of the peptide sequence or membrane lipids explains why no clear

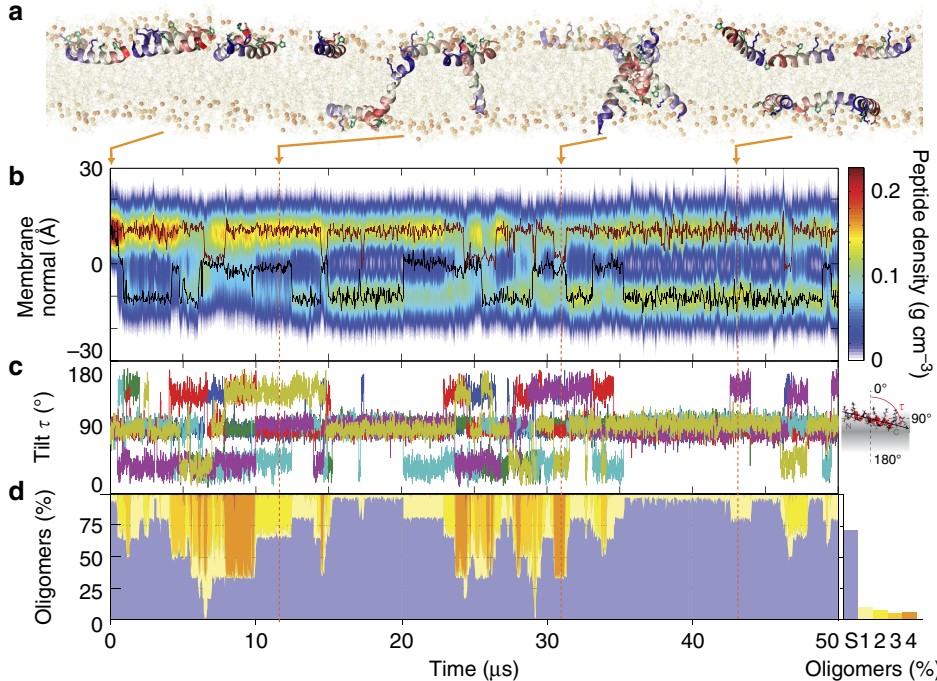

**Figure 3 | Spontaneous membrane insertion and intra-membrane oligomerization of maculatin.** (**a**) All peptides initially reside in the surface-bound state in the upper membrane leaflet (ribbons coloured blue to red from N to C terminus, charged and polar side chains shown). (**b**) Over the course of 50 μs, the peptides first insert TM, translocate to the other bilayer leaflet and then form various oligomers, resulting in an equal distribution of the peptide mass across the membrane. This process is completed after ~15 μs. The centre of mass positions of two representative peptides are shown in red and black. (**c**) Tilt angles of all peptides, showing their numerous transitions between surface-bound and TM-inserted states. (**d**) Dimers, trimers and tetramers both form and dissolve rapidly on a microsecond scale. Monomeric surface-bound states dominate (S, blue), but oligomers, both parallel and antiparallel, occur 29% of the simulation time for this simulation, in all other simulations the oligomer population is 47% (see below). The TM monomer is show in light yellow and higher colour oligomers are shown in darker tones of yellow to orange. A representative antiparallel tetramer is shown at 32 μs.

pore-forming motif has been found in AMP sequences and why bacterial resistance against AMPs is remarkably low despite continuous exposure over millions of years.

**Pore assembly processes**. Pores form and dissipate many times over the course of the simulations, providing a converged description of the assembly mechanisms (Fig. 6). Current models of AMP activity propose detergent-like mechanisms of poration, where interfacial peptide aggregates form pores by concerted insertion. Contrary to these models, we found that the assembly of maculatin into pores is driven by the one-by-one, C-terminal-first addition of surface-bound peptides to an existing TM helix or oligomer. In this way, the translocation of the charged and polar side chains of the inserting peptide are catalysed by the polar face of the already TM-inserted peptide. Direct oligomerization of TM helices is observed only infrequently, and never for large oligomers ($\geq 5$). The pore-forming process appears similar to the spontaneous assembly of integral membrane proteins from fragments[21–23], which is thought to mimic the assembly of multi-span membrane proteins after insertion of the correctly threaded TM helices by the translocon[24].

**Pore properties and relation to AMP activity**. Each pore formed in the assembly simulations is only marginally stable and has its own characteristic architecture, oligomeric state, conductance properties, formation probability and lifetime in the bilayer. Since the typical lifetime of functional pores at this temperature ($<1$ μs) is much shorter than the average assembly time (multi-μs), only a small number of conduction events are typically captured before the pore disassembles again, too few to accurately determine ion selectivity and conductance properties.

To quantify these properties we extracted the most populated pores with long lifetimes from the assembly simulations, equilibrated them in POPC bilayers and performed microsecond timescale spontaneous conductance simulations in the presence of applied voltages. Figure 7 shows the equilibrium ion and water conductance for each type of pore obtained from these simulations (Supplementary Table 2). This shows that conduction of water and ions is dominated by large oligomers with more than five peptides, but lower oligomers can occasionally also leak ions (Supplementary Fig. 2).

To quantify the pore lifetimes at room temperature, we determined the average unfolding times of each pore at highly elevated temperatures (167–207 °C). Unfolding times were found to follow Arrhenius kinetics (Fig. 8). Extrapolation to body temperature (37 °C) gives average pore lifetimes of 24 ms for the octamer, 22 ms for the heptamer and 58 ms for the hexamer.

## Discussion
In summary, experimentally guided simulations revealed that maculatin peptides form ensembles of structurally diverse channel-like pores in the membrane. The pores have a variety of different assembly mechanisms, conductances and lifetimes in the bilayer. Minor mutations or changes in lipid tail length result in different structures and ensembles, which explains AMP resilance against bacterial resistance. Thus, AMPs prevail by fine-tuned interactions that balance pore stability, structural versatility and conductance efficiency.

## Methods
**Methods summary.** In brief, solid-phase synthesized and purified maculatin peptides and mutants were validated using high-pressure liquid chromatography (HPLC) mass spectroscopy. Peptide secondary structure and thermostability was

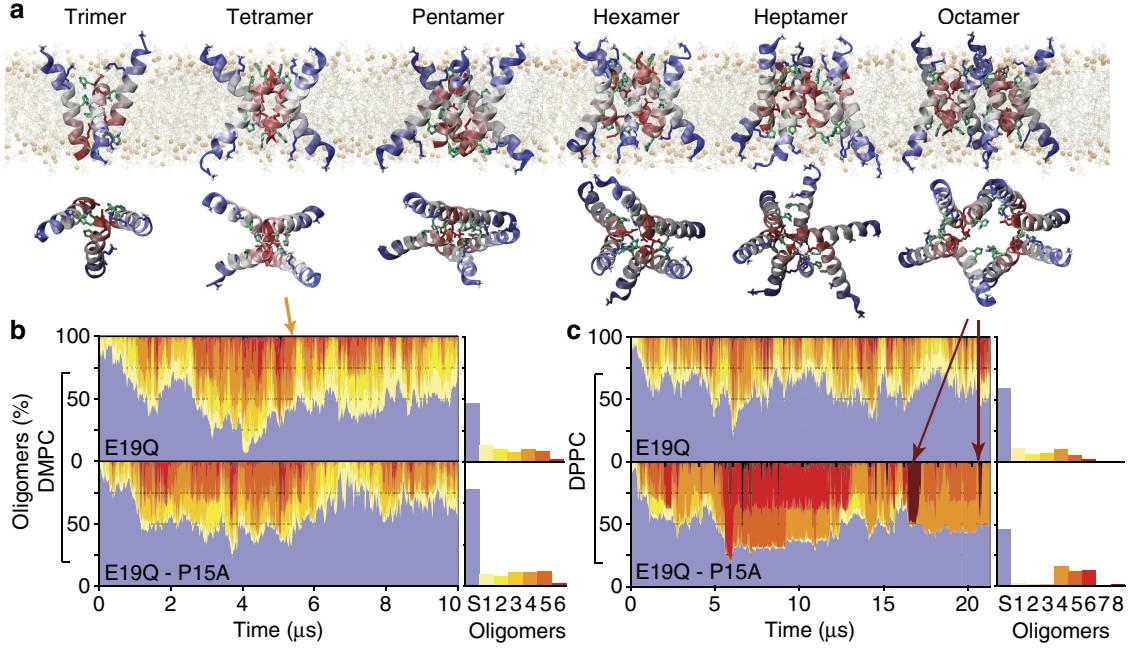

**Figure 4 | Spontaneous formation of an ensemble of structurally diverse channel-like pores.** (**a**) Maculatin forms a large and heterogeneous variety of TM oligomeric structures. The most populated clusters (dimer–octamer) occurring in the simulations are illustrated. (**b**) Oligomer populations vary over time, and the overall occurrence of each oligomer is shown on the right. Antiparallel peptide arrangements are greatly favoured over parallel assemblies and are stabilized by a loosely packed interface made up of the C-terminal polar His12, Gln19 and His20 residues. No higher-order pores than octamers are observed; aggregates of more than eight peptides were found to consist of lower order pores in lateral contact, and occur very infrequently. All simulations show slightly different populations of the lower-order pores ($\leq$pentamer), which usually have dry pores. A thin bilayer (1,2-dimyristoyl-*sn*-glycero-3-phosphocholine; DMPC) cannot support more than a hexamer due to excessive helix tilting. (**c**) Thicker bilayers (1,2-dipalmitoyl-*sn*-glycero-3-phosphocholine; DPPC) allow formation of higher-order pores ($\geq$heptamer), including the water-filled octamer.

determined using synchrotron radiation CD (SRCD) spectroscopy of peptides after addition of large unilamellar vesicles (LUVs) at various peptide to lipid ratios and temperatures. Peptide-binding free energies were determined via vesicle titration, and the average membrane orientation of maculatin and mutant peptides was determined using SRCD spectroscopy of peptides in embedded into oriented lipid bilayer stacks. Peptide-induced pore formation in lipid bilayers was determined by measuring the leakage of dyes from LUVs using fluorescence microscopy. Atomic detail unbiased molecular dynamics simulations were performed using GROMACS (www.gromacs.org) and analysed using HIPPO (www.biowerkzeug.com), using the CHARMM force field.

**Sample preparations.** Maculatin 1.1 and two mutants (P15A and P15A + E19Q) were synthesized and purified by GenScript using Fmoc chemistry, with 98% purity. Peptide purity and identity were confirmed by reverse phase HPLC and electrospray ionization mass spectrometry with the atomic mass of maculatin 1.1 wild-type, P15A and P15A + E19Q are 2145.55, 2119.51 and 2118.53 g mol$^{-1}$, respectively. Peptides were dissolved in distilled water, and the concentration was determined using phenylalanine absorption at 257 nm, assuming an average molar extinction coefficient of $\varepsilon = 200\,M^{-1}\,cm^{-1}$. POPC and other lipids were purchased from Avanti Polar Lipids (Alabaster, AL, USA). All other solvents and reagents were all purchased from Sigma-Aldrich.

**LUV preparation.** A unit of 25 mg lipids (1 ml of 25 mg ml$^{-1}$ lipids in chloroform) were dried under nitrogen gas in a glass vial, and the remaining chloroform was removed under vacuum overnight. Lipids were re-suspended in 10 mM sodium phosphate buffer (pH = 7.0) with 100 mM potassium chloride. The lipids were then extruded 10 times through a 0.1 μm Nucleopore polycarbonate filter to give LUVs of 100 nm diameter. Vesicle size was validated by dynamic light scattering using a Zetasizer Nano ZS.

**ANTS/DPX leakage assay.** A unit of 5 mM ANTS (molecular weight = 427.33 g mol$^{-1}$; Life Technologies) and 12.5 mM DPX (molecular weight = 422.16 g mol$^{-1}$; Life Technologies) were encapsulated in 100 nm POPC LUVs. LUV size and homogeneity was verified by dynamic light scattering using a Zetasizer Nano ZS. Gel filtration chromatography of Sephadex G-100 (GE Healthcare Life Sciences Inc.) was used to remove external free ANTS/DPX from vesicle with entrapped contents. The ANTS/DPX-containing LUVs were diluted to 0.5 mM total lipid concentration. The peptide was added at specific peptide-to-lipid ratios. Leakage

was quantified by the increase in ANTS fluorescence that occurs when the DPX quencher inside the liposome is diluted by leakage into the external buffer on peptide-induced permeabilization of the lipid bilayer. Leakage was measured after 3 h reaction time and measurements were repeated three times. Addition of Triton X-100 (0.05% vol.) to the LUVs, which fully permeabilizes the membrane, served as a positive control.

**Macromolecule release assay.** Several different-size dextrans were prepared and labelled with both TAMRA and biotin. Conjugated dextran was entrapped in POPC LUVs. External dextran was removed by incubation with immobilized streptavidin. Streptavidin labelled with an Alexa-488 fluorophore was added during the leakage experiment, with the peptide as previous described[25]. The sample was incubated for 3 h, before measuring the Alexa-488 fluorescence. A control without added peptide served as the 0% leakage signal, and addition of 0.05% vol. Triton X-100 was used to determine 100% leakage[25].

**CD spectroscopy.** Peptide binding to POPC and 1-palmitoyl-2-oleoylglycero-3-phosphoglycerol (POPG) bilayers was determined by vesicle titration. Peptide solutions (20 μM) in 10 mM phosphate buffer (pH = 7.0) were titrated with 3 μl aliquots of 100 mM POPC/POPG LUVs in identical buffer. CD spectra were recorded using SRCD spectroscopy, using the CD beam-line on ASTRID2 (Aarhus University, Denmark). Spectra were recorded from 270 to 170 nm with a step size of 0.5 nm, bandwidth of 0.5 nm and a dwell time of 2 s. Each spectrum was averaged 3 over repeat scans. The averaged spectra were normalized to molar ellipticity per residue. The raw data were analysed using DichroWeb (http://dichroweb.cryst.bbk.ac.uk/)[26,27].

**Oriented CD spectroscopy.** Peptides were dissolved in chloroform and added to POPC or POPG in chloroform at specific peptide to lipid ratios. Peptide/lipid mixtures were dried under a low flow of nitrogen gas and placed under high vacuum overnight to remove all organic solvent. Dried samples were dissolved in 20 μl of pure 2,2,2-trifluoroethanol, which was spread on a quartz glass slide to form oriented bilayer stacks. After drying the peptide–lipid film under high vacuum to remove all TFE, 2 μl distilled H$_2$O was added to the glass plate, and the plate was placed in a chamber containing a saturated K$_2$SO$_4$ solution (120 g l$^{-1}$ at 25 °C) and equilibrated overnight at 42 °C. CD spectra were recorded using the CD beam-line on ASTRID2 (Aarhus University). Spectra were recorded from 270 to 160 nm with a step size of 0.5 nm, bandwidth of 0.5 nm and a dwell time of 2 s, and

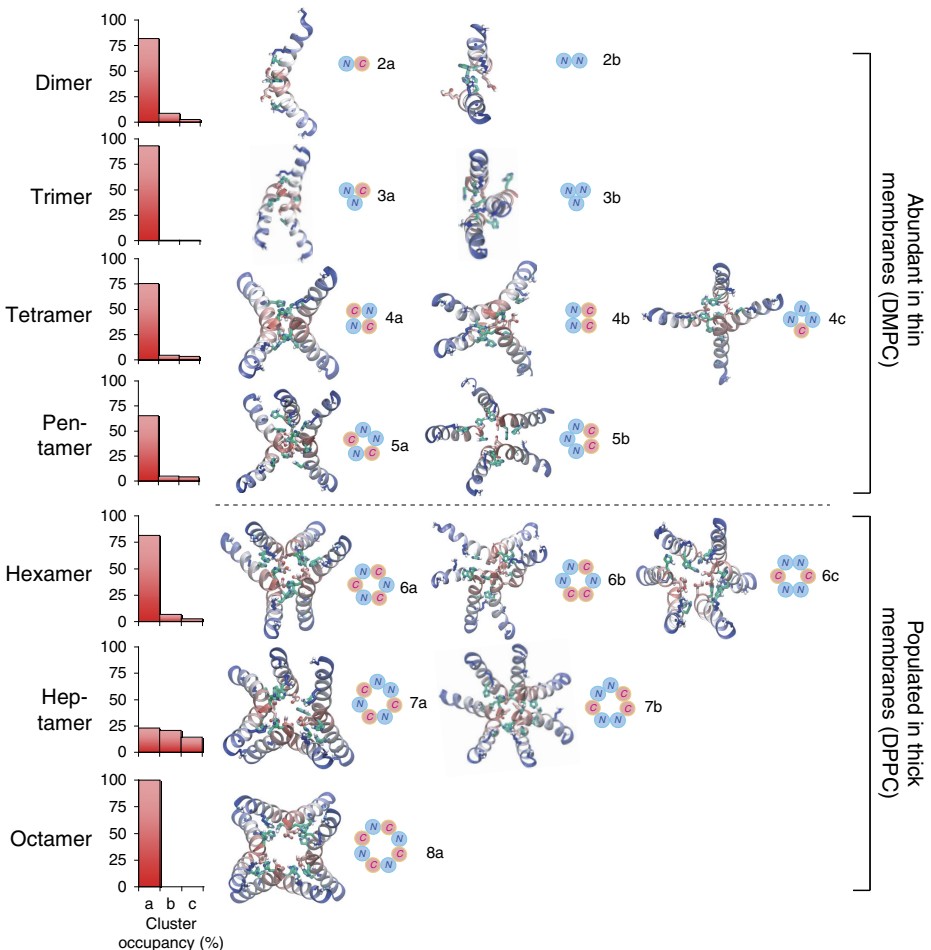

**Figure 5 | Pore ensemble structure variation.** Oligomeric channel-like structures observed in the assembly simulations, together with a corresponding top-view assembly scheme show the versatility of the structural arrangement of the peptides (N: N terminus up, C: C terminus up). The symmetric antiparallel arrangements, assembled from antiparallel peptide pairs (dimer 2a is the basic building block) are always the most populated cluster for each oligomer (cluster 'a'). Membrane thickness can markedly influence the architecture of the TM oligomers formed by maculatin. In thin membranes (for example, 1,2-dimyristoyl-*sn*-glycero-3-phosphocholine; DMPC) only oligomers with a small number of peptides are formed. Oligomers with more than five peptides are found only in thicker membranes (1,2-dipalmitoyl-*sn*-glycero-3-phosphocholine; DPPC), where peptides tilt less, with the octamer forming the largest pore.

averaged over 8 rotational angles, which rotated the sample around the beam axis by 360°. Each spectrum was averaged over 3 repeat scans. The averaged spectra were normalized to molar ellipticity per residue and the results were analysed following the protocols of Huang and Ulrich[28,29].

**Peptide thermostability assay.** SRCD spectroscopy were carried out using the CD beam-line on ASTRID2 (ISA, Aarhus University). Peptides ($20 \mu M$) were embedded in POPC LUV using co-extrusion as described above and solvated in (peptide-to-lipid ratio = 1:50) in 10 mM phosphate buffer (pH = 7.0) were placed in a 1 mm path length quartz cuvette (Hellma Analytics). Spectra were recorded using 0.5 nm bandwidth, 0.5 nm resolution and 2 s dwell time. Temperature was controlled with a Peltier device, and varied between 20 and 90 °C in 10° steps. The actual temperature in the cell was determined using a thermometer and showed a linear correlation with $T_{Actual} = 0.766 \, T_{Setting} + 4.715$ ($R^2 = 0.9999$). The sample was equilibrated for 10 min at each temperature before collecting data, and measurements were repeated five times. The averaged spectra were smoothed using a Fourier filter and normalized to molar ellipticity per residue.

**Binding free-energy assay.** The free energy of folding-partitioning of peptides onto lipid bilayers was determined via vesicle titration. A unit of $200 \mu M$ peptide in 10 mM phosphate buffer (pH = 7.0) was titrated by adding 200, 400, 4,000 and $10,000 \mu M$ POPC LUVs in phosphate buffer at 25 °C. The POPC LUVs had a diameter of 100 nm (determined using dynamic light scattering) and were made via extrusion. Folding-partitioning was determined from the CD signal change at 222 nm on LUV titration. This allows calculation of the binding free energy, as described by Ladokhin *et al.* using the formula $\Delta G = -RT \ln K_X$, where $K_X$ is the partition coefficient. $K_X$ is calculated from the mean residue ellipticity at 222 nm,

$[\theta]_{\lambda = 222nm}$ by assuming that the signal is composed of two states: folded peptide (F) and unfolded peptide (U), which gives $K_X = [F]/[U]$ (refs 30,31).

**Molecular dynamics simulations.** All simulations were performed and analysed using GROMACS version 5.0.5 (www.gromacs.org)[32] and HIPPO beta (www.biowerkzeug.com), using the CHARMM27 force field[33], and TIP3P water[34]. CHARMM36 all-atom lipids parameters were used[35]. Electrostatic interactions were computed using particle-mesh-Ewald, and a cutoff of 10 Å was used for the van der Waals interactions. Bonds involving hydrogen atoms were restrained using LINCS[36]. Simulations were run with a 2 fs time-step, and neighbour lists were updated every five steps. All simulations were performed in the NPT ensemble (constant number of particles, constant pressure and temperature), with water, lipids and the protein coupled separately to a heat bath with $T = 80–120$ °C and a time constant $\tau_T = 0.1$ ps using the velocity-rescaling method. Atmospheric pressure of 1 bar was maintained using the Parrinello − Rahman semi-isotropic pressure coupling method with compressibility $\kappa_z = \kappa_{xy} = 4.6 \times 10^{-5}$ per bar and time constant $\tau_P = 1$ ps.

The high-temperature simulation protocol is robust: In particular, we have demonstrated that elevated temperature simulations of peptide folding-partitioning into lipid bilayers accurately reproduce the native-state structural equilibria and partitioning free energy, compared with both CD and translocon experiments, as well as low temperature simulations[13,37–41]. This approach exploits the fact that the cost of breaking a helical hydrogen bond in the bilayer is very high (~5 kcal mol$^{-1}$), which is much higher than kT, ensuring that membrane-inserted peptides stay fully helical even at highly elevated temperatures[13,37,38].

The initial peptide set-up was the surface bound conformation. This stable interfacial α-helical minimum can be rapidly predicted[39]. Initial simulations were started with all peptides in the same membrane leaflet. These simulations revealed

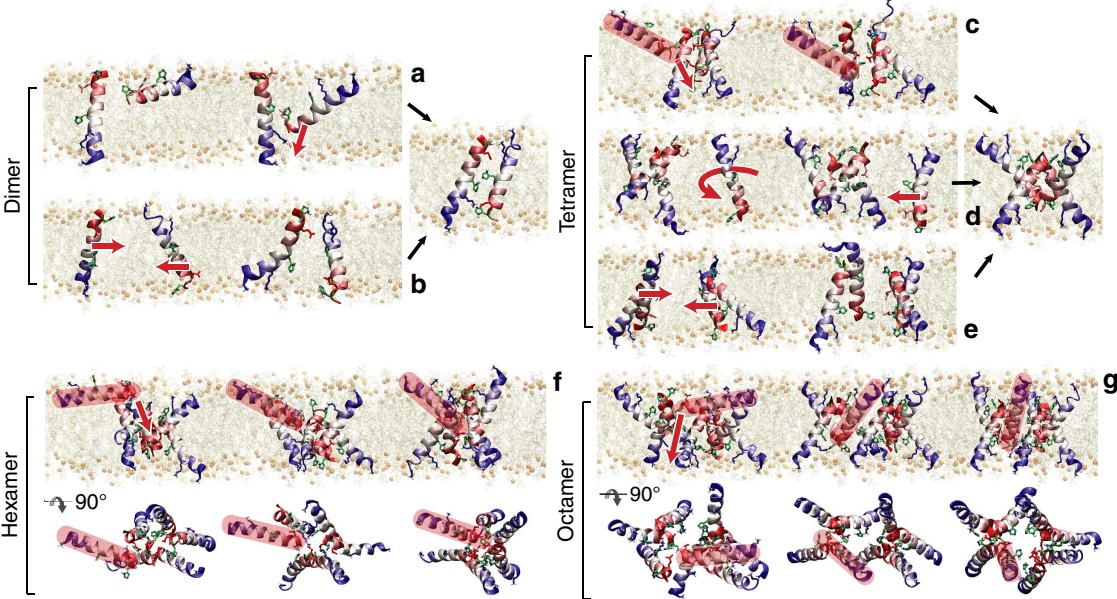

**Figure 6 | Pore assembly process.** (**a**) The dominating assembly process of maculatin 1.1 is a C-terminal-first surface insertion (surface scavenging) catalysed by existing TM helices, which facilitates translocation of the polar side chains. (**b**) In addition, dimers can also form by direct TM–TM oligomerization. (**c**) Similarly, tetramers usually form from a TM trimer via surface recruitment. (**d**) Alternatively, assembly via TM oligomerization requires the joining TM peptide to first rotate to point its polar interface towards the existing trimer. (**e**) In rare cases, an association of two antiparallel dimers can occur. The final tetramer is the same in all three cases. (**f**) Hexamer and (**g**) octamer. Higher-order oligomers assemble solely via surface peptide scavenging (recruited peptide shown in red).

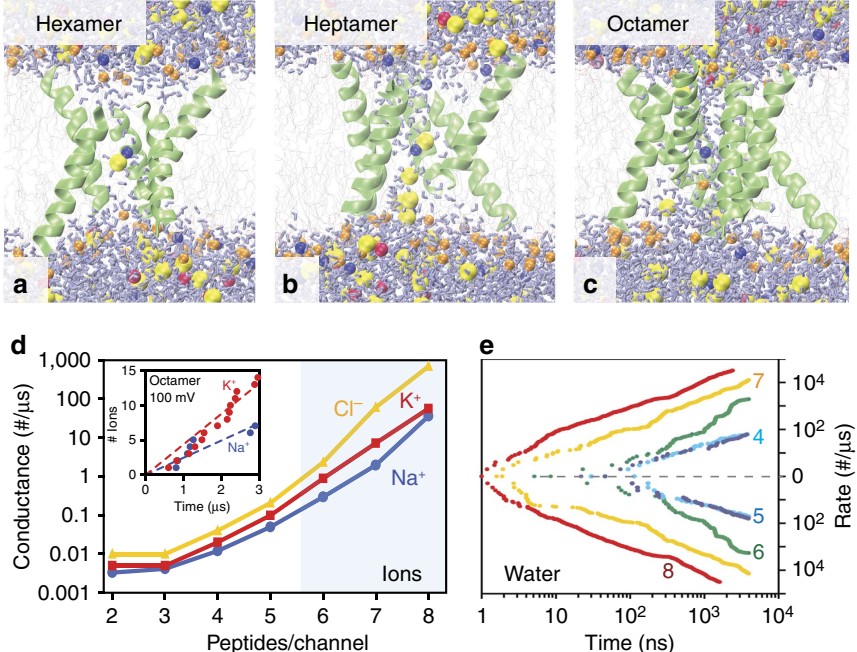

**Figure 7 | Pore selectivities and conductivities.** (**a–c**) Higher-order oligomeric channel-like structures (hexamers, heptamers and octamers) were taken from the pore assembly simulation and equilibrated in a separate lipid bilayer. Water flow and conductance of Na, K and Cl ions were measured from simulations with applied voltages. No restraints were required and the pores remained stable over the 4 μs timescale of the simulations (Supplementary Table 2). (**d**) Conduction and (**e**) water flux are dominated by the large higher-order oligomeric pores (that is, >5 peptides), which were found to be consistent in pore size with the macromolecule vesicle leakage pore-sizing assay (Fig. 1). Pores show a preference for conducting anions (Cl⁻) over cations, and K⁺ conducts roughly 2–3 times the rate of Na⁺.

the long timescales needed for assembly of oligomers that require the equilibration of peptide mass in both leaflets. To speed up the pore-formation process, all subsequent simulations were started with both bilayer leaflets being initially populated. Typical bilayers consisted of 200 14:0 phosphatidylcholine:1,

2-dimyristoyl-*sn*-glycero-3-phosphocholine, 16:0 phosphatidylcholine:1, 2-dipalmitoyl-*sn*-glycero-3-phosphocholine or 18:0 phosphatidylcholine:1, 2-distearoyl-*sn*-glycero-3-phosphocholine lipids, and ∼30 water molecules per lipids[39]. All simulations were performed on the PI supercomputer at Shanghai

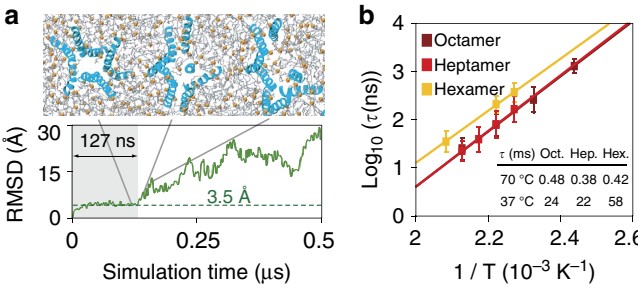

**Figure 8 | Pore lifetimes.** Pore lifetimes at body temperature (37 °C) were estimated using high-temperature unfolding simulations. (**a**) Pores were taken from the assembly simulation (Fig. 3) and equilibrated in a separate bilayer. Subsequently, the equilibrium-unfolding times ($\tau$) were measured at elevated temperatures by calculating the pore r.m.s.d. with respect to the initial structure. Pore-unfolding events (here pore **a**) can be identified by rapid rises in r.m.s.d. from an equilibrated baseline (3.5 Å), which is generally associated with pore dissociation. (**b**) The temperature dependence of pore-unfolding times shows perfect Arrhenius behaviour, allowing extrapolation to lower temperatures. At body temperature this gives typical pore lifetimes of tens of milliseconds. Each point in this graph is an average of eight simulations (Supplementary Table 3). Error bars are s.e.m.

Jiao-Tong University and the MARCC supercomputing facility at Johns Hopkins University. Typical simulation length was between 20–50 μs for each trajectory, sufficient to converge the oligomer populations. For all quantities, an estimate of the s.e.m. was calculated by automatically block averaging over 10 blocks, and these are plotted as error bars.

**Conduction simulations and pore lifetime simulations.** Pore structures were taken from the assembly simulations and equilibrated for 100 ns in POPC bilayers and 500 mM Na/K/Cl solution, while holding the pore structure in place using positional restraints on the backbone carbon atoms. After equilibration of the membrane around the pore restraints were removed and spontaneous conduction was measured by counting the transport events of ions and water through the pore in the presence of applied voltages (Supplementary Table 2). Functional pore lifetimes were determined by repeating the simulations at increasing temperatures and recording the dissolution times of the pore (Supplementary Fig. 7).

**Analysis.** *Peptide density.* The equilibration of the peptide mass from one leaflet across the membrane was calculated by performing a histogram of the peptide mass as a function of the membrane normal and simulation time. The histogram was smoothed over 10 consecutive trajectory frames.

*Oligomer population.* To reveal the most populated pore assemblies during the simulations, a complete list of all oligomers was constructed for each trajectory frame. An oligomer of order $n$ was considered any set of $n$ peptides that are in mutual contact, defined as a heavy-atom (N, C and O) minimum distance of $<3.5$ Å. Population plots of the occupation percentage of oligomer $n$ multiplied by its number of peptides $n$ were then constructed. These reveal how much peptide mass was concentrated in which oligomeric state during the simulation time.

*Permutational cluster analysis.* The enormous amount of heterogeneous structures formed by maculatin 1.1 over the course of the simulations posed a significant challenge for automated analysis. Conventional cluster analysis could not be used, as the same specific oligomeric pore could consist of entirely different peptides in each instance. To address this, we have developed a new permutational cluster analysis tool as part of the HIPPO analysis programme. After the oligomer list was computed, in a second step, all oligomers of the same order $n$ were conformationally clustered. A clustering algorithm was used, with a backbone root mean squared deviation (r.m.s.d.) similarity cutoff criterion of 4 Å. Since each oligomer could be made up of different peptides—or of the same peptides but in a different order—the clustering had to compare one oligomer with all $n!$ permutations of peptide arrangements of the other oligomer. Permutations were generated using Heap's algorithm. For large $n$, these computations could last several hours. The final r.m.s.d. value of the conformational similarity was considered the lowest r.m.s.d. value as obtained from the $n!$ permutational comparisons.

**Data availability.** The data that support the findings of this study are available from the corresponding author on reasonable request.

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

## Acknowledgements

We thank the Aarhus synchrotron CD beamline staff for support and beamtime (Grant #ISA-16-1038, to M.B.U.), and Johns Hopkins University for funding. This research was supported by a China 1000 Plan's Program for Young Talents (#13Z127060001) to J.P.U. and by National Natural Science Foundation of China (NSFC) grants #91230202 and #11471213 to D.H. Simulation resources were supported by Center for High Performance Computing, Shanghai Jiao Tong University and the MARCC supercomputer facility at Johns Hopkins University.

## Author contributions

Y.W., D.H., M.B.U. and J.P.U. designed the research; Y.W. performed the simulations; C.H.C. performed the experiments; Y.W., M.B.U. and J.P.U. performed molecular dynamics analyses; M.B.U. and J.P.U. developed software and analysed the data; Y.W., M.B.U. and J.P.U. wrote the paper with input from the other authors.

## Additional information

**Competing financial interests:** The authors declare no competing financial interests.

**Publisher's note**: 

