## [Peer Review File · Nature Communications]

Reviewer #1 (Remarks to the Author)

This paper by Ulmschneider and collaborators presents a very careful and molecular dynamics (MD) simulations of the interaction of the amphipathic peptide maculatin with phospholipid membranes. The MD simulations are rendered the more relevant because of the close connection to experiments, also performed in this work, which guide the simulations to reproduce the experimentally known properties of the peptide. This lends additional confidence to the simulation results. The simulations use the most accurate (probably the only accurate) force field available for this type of study (CHARMM).

The authors find that the peptide forms a plethora of oligomer structures, whose relative populations depend on peptide concentration and membrane type (namely, bilayer thickness). This is consistent with experimental observations known for a number of amphipathic (mostly antimicrobial or cytolytic) peptides, whose mechanism on membranes---that is, whether it forms pores, slowly perturbs the membrane, or causes some catastrophic bilayer breakdown---seems to vary significantly depending on peptide/lipid concentration and on lipid composition. This study is illuminating regarding why experimentalists should perhaps stop discussing what is the mechanism of a certain peptide: the question may be rather meaningless in general. In the case of maculatin, the simulations show a predominance of dimers at low peptide concentration, and a predominance of higher oligomers (hexamers) at high concentration (again dependent on the lipid used). Perhaps one of the most interesting aspect of the simulations is that initial insertion into the bilayer is dominated by monomers or dimers, which work as catalysts for the insertion of other monomers and as seeds for the assembly of larger pores. Another very important aspect is identification in the simulations of a specific dimer, a head-to-tail well-defined structure. This is important because it is a testable prediction. (As far as I know this dimer structure has not been determined experimentally by NMR.) Its experimental validation is now a challenge for NMR spectroscopists, well within reach of current NMR methods. If confirmed, this dimer structure would be a powerful argument for the validity of the MD simulations. For these reasons, I think this paper is highly significant.

Minor comments and questions:

- 1) Figure 1 legend: the word "colour," in the second-to-last sentence, should be deleted.
- 2) I understand that the Gibbs energy of binding reported (-4.6 kcal/mol, page 2 and Extended Date Figure 2) is calculated from the experimental equilibrium binding constant determined by CD titration. However, what is the standard state? Is this Gibbs energy based on a molar or on a mole fraction concentration scale? The two scales give Gibbs energies that differ by 2.4 kcal/mol at room temperature ($RT \ln[\text{Water}]$), so it is important to know which was used in case for comparison between MD simulations and experiment.
- 3) Can an estimate be provided for the Gibbs energy of peptide insertion of a monomer (and a dimer), from the membrane interface to the bilayer interior?
- 4) On page 3 (middle), the sentence "This shows that while maculatin forms proper channels in the membrane, rather than disordered pores or detergent-like holes, the channel equilibrium can be easily perturbed." is unclear to me. First I don't see why "This," which refers to the previous statement, shows what the authors claim it does. Perhaps my difficulty is with what is meant by "easily perturbed." Second, I am not sure the premise is correct: can you really say that maculatin forms proper channels? Doesn't the whole paper show that the type of structure formed depends on the peptide concentration and on the lipid used? Please clarify.

Reviewer #2 (Remarks to the Author)

Wang et al. use a long timescale molecular dynamics (MD) to describe how an antimicrobial

peptide, maculatin 1.1, acts in model phospholipid membranes. The MD results are compared to CD spectroscopy and dye leakage studies. It appears that maculatin forms a transmembrane α -helix and acts via pore-formation, with up to 8 peptides lining the lumen. These pores are large enough to pass ions, water and small dyes. Interestingly the authors call these channels, which is a semantic point. Channels usually conduct a certain species (e.g., ions) and tend to be selective, so that maculatin could be said to form small pores with channel-like structure. There are a few small details listed below that the authors should address:

Experimental set-up, p2: Gram is a person's name and should be capitalized.

Insertion, p2: What current models propose insertion of large aggregates - please give a reference as usually small oligomers are suggested?

Spontaneous formation, p3: The barrel-stave and toroidal pores could also be heterogeneous, so would not preclude such models. Possibly, a qualified could be inserted, e.g., 'such as generally assumed'.

Contrarily, based on F15 in SI (p16), the WT peptide is more effective in inducing leakage in some lipids (e.g., 16:1), so statement about little/no difference could be softened

Could the authors comment on the antibacterial activity of the mutant peptides - is it similar to the wild type?

Channel selectivity, p4: Did the authors measure ion conductance experimentally to compare to the simulations in Fig 4?

Summary, p4: Possibly use the words 'channel-like pores' here to better convey the mechanism?

Future MD study could address anionic phospholipids as model bacterial membranes.

Methods, p10: Give peptide purity and atomic mass.

p11: Note that for the oriented CD spectra, the lipids are not fully hydrated and peptides would behave differently to when excess water is present.

Is the POPC stable at 90 deg C? Care should be taken when heating unsaturated lipids.

p12: Interesting that simulations were done with saturated lipids. Is bilayer thickness for POPC similar to DMPC? See Sani et al. (2012) *Biochim Biophys Acta* 1818:205 for CD study of maculatin in different chain length lipid.

p17: What does 100% hydration mean in caption to Fig 2 SI? There would be no excess water at this relative humidity, but less than 30 waters per PC which could be considered fully hydrated based on lipid dynamics as measured by solid-state NMR. See Kuo & Wade (1979) *Biochemistry* 18:2300, and Elworthy, P.H. (1961) *J. Chem. Soc.* 5385.

Reviewer #3 (Remarks to the Author)

The manuscript describes a combination of molecular dynamics simulations and experiments designed to explain the process by which maculatin, an antimicrobial peptide, porates membranes. The main result is that there is not a single pore structure, but rather an ensemble of structures, with a range of sizes and conductivities. This probably shouldn't be surprising, but it is contrary to the models often invoked in the literature, and the results are quite striking and offer important insights.

The simulations are extremely long by current standards -- the only comparable during simulations are from the DE Shaw group, who haven't considered a system of this kind -- which in and of itself makes the work impressive. I have some technical qualms with the way the simulations were run (see below), but on the whole the analysis is excellent, and the manuscript is clearly written and explained.

Methodological complaints/questions:

-- Why did the authors choose to use the Berendsen thermostat and barostat?

This is a technique known to produce incorrect thermodynamic ensembles (which the authors clearly know, since they compensate by using separate thermostats for the lipids, water, and peptides), and Gromacs has implemented correct methods for years. Most likely, this choice did not alter any of the conclusions, but it's very frustrating to see this continued use of sub-standard methods.

-- The stated way of computing error bars is simply incorrect, in two different ways. First, the authors call it "block averaging", which properly refers to the technique from Flyvbjerg and Jenson (JCP, 1991, I think), which is not what the authors did. Second, choosing to break the system into an arbitrary number of blocks allows the authors to effectively choose what their error bars look like. I suspect there's enough data to do the block averaging correctly (measure the block standard error as a function of block size and find the plateau value), but if not the authors should simply say so. To my mind, the specific computed numbers aren't nearly as important as the overall message, which would not be invalidated by a lack of error bars.

-- Simulating above the boiling temperature of water is at best a disconcerting and at worst incorrect method of speeding the sampling. Essentially, the authors are relying on the fact that phase transitions like evaporation occur on too slow a timescale to show up in the simulation, but the fact is that the systems are out of equilibrium.

-- PC is an odd choice of lipid headgroup, since it is entirely absent in bacteria and lacks the charge and negative intrinsic curvature expected for more common bacterial lipids like PE and PG. The experiments are done in PC as well, so it's a fair comparison, but this is not a lipid that resembles the target membranes.

-- Do you think the results about relative populations of clusters with particular mixtures of head-to-tail peptides are effected by starting with some peptides on each leaflet (given that dropping the N-terminus through the membrane appears to be harder than the C-terminus)? Similarly, do you think having a biological transmembrane voltage would alter the results?

Minor issues:

-- page 12, "these computations could be heavy" is sloppy writing. Please clarify.

-- Extended Data Figure 2. The caption for part B doesn't say what the red and blue curves are.

-- I'm confused by the way the setup is described. In discussing the setup on page 2 the authors state that the peptides "were initially placed on one interface". However, on page 12 the authors make it clear that that's only true for the systems with 6 peptides, and that the other systems were constructed symmetrically. This needs to be clearer in the main body of the text.

-- It might be nice to see reference to the experimental work of folks like Bill Wimley, Heiko Heerklotz, or Paulo Almeida, who've talked about all-or-nothing vs. graded leakage mechanisms, which seems connected to the main conclusions of this manuscript.

Reviewers' comments:

Reviewer #1 (Remarks to the Author):

This paper by Ulmschneider and collaborators presents a very careful and molecular dynamics (MD) simulations of the interaction of the amphipathic peptide maculatin with phospholipid membranes. The MD simulations are rendered the more relevant because of the close connection to experiments, also performed in this work, which guide the simulations to reproduce the experimentally known properties of the peptide. This lends additional confidence to the simulation results. The simulations use the most accurate (probably the only accurate) force field available for this type of study (CHARMM).

The authors find that the peptide forms a plethora of oligomer structures, whose relative populations depend on peptide concentration and membrane type (namely , bilayer thickness). This is consistent with experimental observations known for a number of amphipathic (mostly antimicrobial or cytolytic) peptides, whose mechanism on membranes---that is, whether it forms pores, slowly perturbs the membrane, or causes some catastrophic bilayer breakdown---seems to vary significantly depending on peptide/lipid concentration and on lipid composition. This study is illuminating regarding why experimentalists should perhaps stop discussing what is the mechanism of a certain peptide: the question may be rather meaningless in general. In the case of maculatin, the simulations show a predominance of dimers at low peptide concentration, and a predominance of higher oligomers (hexamers) at high concentration (again dependent on the lipid used). Perhaps one of the most interesting aspect of the simulations is that initial insertion into the bilayer is dominated by monomers or dimers, which work as catalysts for the insertion of other monomers and as seeds for the assembly of larger pores. Another very important aspect is identification in the simulations of a specific dimer, a head-to-tail well-defined structure. This is important because it is a testable prediction. (As far as I know this dimer structure has not been determined experimentally by NMR.) Its experimental validation is now a challenge for NMR spectroscopists, well within reach of current NMR methods. If confirmed, this dimer structure would be a powerful argument for the validity of the MD simulations. For these reasons, I think this paper is highly significant.

Minor comments and questions:

1) Figure 1 legend: the word "colour," in the second-to-last sentence, should be deleted.

We have corrected this.

2) I understand that the Gibbs energy of binding reported (-4.6 kcal/mol, page 2 and Extended Data Figure 2) is calculated from the experimental equilibrium binding constant determined by CD titration. However, what is the standard state? Is this Gibbs energy based on a molar or on a mole fraction concentration scale? The two scales give Gibbs energies that differ by 2.4 kcal/mol at room temperature ($RT \ln[Water]$), so it is important to know which was used in case for comparison between MD simulations and experiment.

The two structural states are coil in aqueous solution and helical structure in the more hydrophobic membrane. Therefore, the standard states of the free energy are \$G^{\circ}_{solution}\$ and \$G^{\circ}_{membrane}\$. However, the peptide in the membrane may have more states that contribute to the CD signal in the vesicle titration, e.g. surface bound peptides, transmembrane inserted peptides, and peptide oligomers. In our calculation, we use the molar scale for the Gibbs free energy and the equilibrium

binding constant is determined using $K_x = ([P]_{\text{bilayer}}/([L]+[P]_{\text{bilayer}}))/([P]_{\text{water}}/([W]+[P]_{\text{water}}))$.

3) Can an estimate be provided for the Gibbs energy of peptide insertion of a monomer (and a dimer), from the membrane interface to the bilayer interior?

Yes, this is simply the logarithm of the populations given in the histogram in Figure 2. $\Delta G_{S \rightarrow TM} = -kT \ln (P(S)/P(TM)) = 0.1$ kcal/mol averaged over all 4 simulations (P(S) = surface bound population, P(TM) = transmembrane inserted population), with TM denoting any TM oligomer. Similarly low free energy differences are seen between the monomer and the dimer. We have added the $\Delta G_{S \rightarrow TM}$ number to the text.

4) On page 3 (middle), the sentence "This shows that while maculatin forms proper channels in the membrane, rather than disordered pores or detergent-like holes, the channel equilibrium can be easily perturbed." is unclear to me. First I don't see why "This," which refers to the previous statement, shows what the authors claim it does. Perhaps my difficulty is with what is meant by "easily perturbed." Second, I am not sure the premise is correct: can you really say that maculatin forms proper channels? Doesn't the whole paper show that the type of structure formed depends on the peptide concentration and on the lipid used? Please clarify.

We have modified this section to clarify the meaning: The main populated 'channels' are of a stable, specific structure (rather than disordered pores), but their populations are easily changed via mutation or different membrane thickness.

We thank you for reviewing our manuscript!

Reviewer #2 (Remarks to the Author):

Wang et al. use a long timescale molecular dynamics (MD) to describe how an antimicrobial peptide, maculatin 1.1, acts in model phospholipid membranes. The MD results are compared to CD spectroscopy and dye leakage studies. It appears that maculatin forms a transmembrane α -helix and acts via pore-formation, with up to 8 peptides lining the lumen. These pores are large enough to pass ions, water and small dyes. Interestingly the authors call these channels, which is a semantic point. Channels usually conduct a certain species (e.g., ions) and tend to be selective, so that maculatin could be said to form small pores with channel-like structure. There are a few small details listed below that the authors should address:

Experimental set-up, p2: Gram is a person's name and should be capitalized.

We have corrected this.

Insertion, p2: What current models propose insertion of large aggregates - please give a reference as usually small oligomers are suggested?

This is just one of the many models proposed for AMP activity. For example, S. Marrink and coworkers proposed that peptide aggregation, either prior to or after binding to the membrane surface, is a prerequisite to pore formation for some AMPs (e.g. melittin and magainin). [reference: D. Sengupta, et al. BBA-Biomembrane 2008 Jun; 1778: 2308-17 and L. T. Nguyen, et al. Trends Biotechnol. 2011 sep; 29(9):464-72]. We have added some citations to the text.

Spontaneous formation, p3: The barrel-stave and toroidal pores could also be heterogeneous, so would not preclude such models. Possibly, a qualified could be inserted, e.g., 'such as generally assumed'.

We have adjusted the sentence as suggested. Meant was that none of the generally proposed models of AMP channel formation applies for maculatin.

Contrarily, based on Fi5 in SI (p16), the WT peptide is more effective in inducing leakage in some lipids (e.g., 16:1), so statement about little/no difference could be softened

Maculatin 1.1 WT does indeed induce higher leakage than its mutant, P15A. Thank you for spotting this. We have corrected the statement.

Could the authors comment on the antibacterial activity of the mutant peptides - is it similar to the wild type?

Maculatin 1.1 WT has slightly more efficient antibacterial activity than the mutant P15A (Niidome, et al. J Pept Sci. 2004 Jul;10(7):414-22.). A previous NMR study shows proline at residue 15 can induce a significant change in membrane order and affect the ability of the bilayer to recover from structural changes induced by the binding and insertion of the peptide (Fernandez, et al. Biophys J. 2013 Apr 2; 104(7): 1495–1507.). However, it is not clear what role proline plays in the assembly process of trans-membrane channel pores, or the pore structure itself. Our guess is that the proline may assist in making the peptide more flexible, giving a wider range of pores that promote leakage in the membrane, as well as increase the solubility of the peptide. Intriguingly, how all of this affects antibacterial activity remains unknown.

Channel selectivity, p4: Did the authors measure ion conductance experimentally to compare to the simulations in Fig 4?

This would indeed be exciting data to have and we have recently initiated a collaboration to do this, but it turns out to be much harder than expected and currently we have no results to report. But we hope to overcome the technical problems and include conductance measurements in future work.

Summary, p4: Possibly use the words 'channel-like pores' here to better convey the mechanism?

We have corrected this.

Future MD study could address anionic phospholipids as model bacterial membranes.

We agree and are currently working on this.

Methods, p10: Give peptide purity and atomic mass.

We have added these quantities to the manuscript. The peptide has 98% purity and the atomic mass of maculatin 1.1 WT, P15A, and P15A+E19Q are 2145.55 g/mol, 2119.51 g/mol, and 2118.53 g/mol, respectively.

p11: Note that for the oriented CD spectra, the lipids are not fully hydrated and peptides would behave differently to when excess water is present.

This is correct and a general limitation of oriented CD of peptides in lipid bilayers. We generally check the oriented CD spectrum while hydrating the peptide-lipid film to check for convergence. Thus full hydration here means that the lipid bilayer stack on the quartz slide has hydrated to the maximum possible extent, providing a converged CD spectrum. Hydration is done by placing the films in a chamber at 100% relative humidity, typically for 24-48 hours before collecting data.

Is the POPC stable at 90 deg C? Care should be taken when heating unsaturated lipids.

Good point. We have previously used POPC at up to 95°C for a range of other membrane active peptides and obtained virtually identical spectra as when using DMPC, DPPC, and DOPC (Ulmschneider et al. JACS 132, 3452–3460). This suggests the bilayer is stable. We also routinely cool the sample back down to room-temperature after a heating scan and check for any change in the CD spectrum or opacity of the vesicle suspension.

p12: Interesting that simulations were done with saturated lipids. Is bilayer thickness for POPC similar to DMPC? See Sani et al. (2012) Biochim Biophys Acta 1818:205 for CD study of maculatin in different chain length lipid.

The paper suggests that DMPC has a slightly broader hydrophobic core than POPC. Indeed in our simulations we observe the hydrophobic width to be roughly identical.

p17: What does 100% hydration mean in caption to Fig 2 SI? There would be no excess water at this relative humidity, but less than 30 waters per PC which could be considered fully hydrated based on lipid dynamics as measured by solid-state NMR. See Kuo & Wade (1979) Biochemistry 18:2300, and Elworthy, P.H. (1961) J. Chem. Soc. 5385.

As mentioned above, we generally check the oriented CD spectrum while hydrating the peptide-lipid film to check for convergence. Thus full hydration here means that the lipid bilayer stack on the quartz slide has hydrated to the maximum possible extent, providing a converged CD spectrum. Hydration is done by placing the films in a chamber at 100% relative humidity, typically for 24-48 hours before collecting data. The bilayers do indeed remain fluid, and peptides can partition into and out of them.

We thank you for reviewing our manuscript!

Reviewer #3 (Remarks to the Author):

The manuscript describes a combination of molecular dynamics simulations and experiments designed to explain the process by which maculatin, an antimicrobial peptide, porates membranes. The main result is that there is not a single pore structure, but rather an ensemble of structures, with a range of sizes and conductivities. This probably shouldn't be surprising, but it is contrary to the models often invoked in the literature, and the results are quite striking and offer important insights.

The simulations are extremely long by current standards -- the only

comparable during simulations are from the DE Shaw group, who haven't considered a system of this kind -- which in and of itself makes the work impressive. I have some technical qualms with the way the simulations were run (see below), but on the whole the analysis is excellent, and the manuscript is clearly written and explained.

Methodological complaints/questions:

-- Why did the authors choose to use the Berendsen thermostat and barostat?

This is a technique known to produce incorrect thermodynamic ensembles (which the authors clearly know, since they compensate by using separate thermostats for the lipids, water, and peptides), and Gromacs has implemented correct methods for years. Most likely, this choice did not alter any of the conclusions, but it's very frustrating to see this continued use of sub-standard methods.

Thank you for spotting this! This is an embarrassing and unfortunate error in the method section. We actually have been using the v-rescale and Parrinello-Rahman methods for thermostat and barostat for many years. These methods have been demonstrated to generate a proper canonical ensemble [Bussi, JCP 2007, 126, 014101]. The method section has been corrected.

-- The stated way of computing error bars is simply incorrect, in two different ways. First, the authors call it "block averaging", which properly refers to the technique from Flyvbjerg and Jenson (JCP, 1991, I think), which is not what the authors did. Second, choosing to break the system into an arbitrary number of blocks allows the authors to effectively choose what their error bars look like. I suspect there's enough data to do the block averaging correctly (measure the block standard error as a function of block size and find the plateau value), but if not the authors should simply say so. To my mind, the specific computed numbers aren't nearly as important as the overall message, which would not be invalidated by a lack of error bars.

This is correct. Our analysis tools automatically report an estimate of the standard error of the mean as the standard deviation of the means of a fixed number of blocks with respect to the overall mean. Implicit in this is that each block is longer than the correlation time, so the blocks are statistically independent (i.e. the plateau value as mentioned). The correct way (but not easily doable in a fully automated fashion) of doing this would be to calculate the statistical inefficiency s (= correlation time) first, then divide the simulation into N/s blocks (N =total number of frames), and calculate the standard error of the mean. However, we have programmed fully automated tools as the length and number of simulations makes analysis by hand very time-consuming and error prone, so we use a fixed number of blocks. This can lead to an error estimate that is too small, when the block length greatly exceeds the correlation time. If the blocks are shorter than the correlation time, the error value will also be incorrect. We have made these descriptions clearer in the method section, where we now call the error bars 'estimates'.

-- Simulating above the boiling temperature of water is at best a disconcerting and at worst incorrect method of speeding the sampling.

Essentially, the authors are relying on the fact that phase transitions like evaporation occur on too slow a timescale to show up in the simulation, but the fact is that the systems are out of equilibrium.

Since high-temperature techniques may be criticized, we have developed this approach carefully for many years, always verifying against either experimental results, or against low-temperature simulations that require orders of magnitude longer simulations (e.g. Ulmschneider et al. JACS 2011, 133, 15487–15495, Ulmschneider et al. Nature Comm. 2014, 5, 4863). Several of our prior studies are now cited in the text, where the reliability of the high-temperature technique is illustrated for a variety of synthetic peptides and bilayers. We have made the method section more clear on this. We would also like to point out that while all barostats can capture the solid-liquid phase transitions, none of the MD barostats are able to capture a liquid-to-vapor phase transition. This requires different algorithms that are currently not implemented in the main MD packages. High temperatures are also routinely used in replica exchange (or parallel tempering) simulations, where temperatures as high as 1000 K are sometimes used.

-- PC is an odd choice of lipid headgroup, since it is entirely absent in bacteria and lacks the charge and negative intrinsic curvature expected for more common bacterial lipids like PE and PG. The experiments are done in PC as well, so it's a fair comparison, but this is not a lipid that resembles the target membranes.

This is a good point. We thought about what lipids to choose when designing the experiments, but chose PC lipids in the end since a large body of AMP experiments (e.g. SS-NMR) have been performed with PC (often DMPC) (E. Strandberg et al. BBA 2009, 1788, 1667 – 1679), mimicking mammalian membranes. Testing for PE and PG, or mixtures for bacterial membranes is currently performed. Our initial results for several AMPs suggest that the overall results are similar in mixed PC:PG membranes, but since these simulations take over a year to run it is too early to tell if the results are converged.

-- Do you think the results about relative populations of clusters with particular mixtures of head-to-tail peptides are effected by starting with some peptides on each leaflet (given that dropping the N-terminus through the membrane appears to be harder than the C-terminus)? Similarly, do you think having a biological transmembrane voltage would alter the results?

No, the relative populations of the oligomers at equilibrium is irrespective of the initial conditions. We simply started some simulations with the peptides already on both leaflets as this speeds up the time the system takes to reach equilibrium. We checked this by building simulations that place all peptides on one leaflet of the lipid bilayer (simulations DM1-DM4) to mimic the experimental initial conditions in the vesicle titration assay. In all these systems the peptides translocate through the membrane slowly and the system finally ends up with the peptides symmetrically distributed across the bilayer, and a roughly equal number of peptides on each membrane interface. We believe this is also what happens in the experimental vesicle titration. Since this process is very long (>50 microseconds) we decided to start subsequent simulations with fully

symmetric leaflets, as we were interested to see if we can capture pore formation.

Transmembrane voltages will almost certainly influence the ensemble distributions, by for example altering the ratio of parallel oligomers over antiparallel oligomers. However, since there is no membrane potential in the leakage assays, we did not apply a voltage here, in order to mimic the experimental conditions closely. We are currently performing simulations with applied voltage to investigate the size of the effect on the channel equilibrium.

Minor issues:

-- page 12, "these computations could be heavy" is sloppy writing. Please clarify.

We have rephrased this sentence.

-- Extended Data Figure 2. The caption for part B doesn't say what the red and blue curves are.

These two spectra correspond to theoretical CD spectra for helices aligned perfectly along the beam (red dashed curve) and perpendicular to the beam (dashed blue curve) are shown. This corresponds to transmembrane inserted and surface bound states respectively, and can be used to estimate the net ratio of inserted to surface bound maculatin. We have added the corresponding explanation to the caption.

-- I'm confused by the way the setup is described. In discussing the setup on page 2 the authors state that the peptides "were initially placed on one interface". However, on page 12 the authors make it clear that that's only true for the systems with 6 peptides, and that the other systems were constructed symmetrically. This needs to be clearer in the main body of the text.

We have now made this more clear. The first set of simulations were used to show that peptides placed initially on one interface equilibrate across the bilayer by transitioning through the membrane, resulting ultimately in equal populations of peptides on both interfaces (simulations DM1-DM4). Since this process is very slow (estimate > 50 microseconds), we did not want to directly simulate it for 16 peptides (which is computationally much more costly than the 6 peptide simulation). So after making the point how insertion works, we only considered the equilibrium phase for those later simulations with many more peptides. The rapid formation of pores is only observed in this phase.

-- It might be nice to see reference to the experimental work of folks like Bill Wimley, Heiko Heerklotz, or Paulo Almeida, who've talked about all-or-nothing vs. graded leakage mechanisms, which seems connected to the main conclusions of this manuscript.

We are indeed highly indebted to the large body of work by Bill, Paulo, and Heiko Heerklotz. We have added references to a

previous study on GUV leakage (Frances Separovic, et al. *Biophys. J.*, (2005), 89: 1874-1881), which shows that maculatin shares all-or-none behavior with magainin 2 (Paulo Almeida *Biophys. J.*, (2009), 96(1): 116-131), human defensin 2 (William C. Wimley, et al. *Protein Sci.* (1994), 3: 1362-1373), and Agrastatin 1 (Heiko Heerklotz *BBA-biomembranes*, (2011); 1808 (8): 2000-2008). We have added a sentence and the citations to the text.

We thank you for reviewing our manuscript!

Reviewer #1 (Remarks to the Author)

A. Response to my comments.

The authors have adequately responded to my questions. One minor remark related to my point #2: a molar scale was used to express concentrations in water, but the K_x as defined by the authors is actually a mole fraction partition coefficient (mole fraction of peptide in the bilayer divided by mole fraction in water), not a molar partition coefficient (which would be moles of peptide per liter of bilayer divided by moles of peptide per liter of water). The two coefficients differ by the ratio of the molar volumes of water to lipid. As it is, the Gibbs energies of transfer calculated in the paper should be directly comparable with calculations using the Wimley-White scale.

B. Response to the comments of reviewer #3.

At the request of the editor, I have read the authors' response to reviewer #3. In my opinion, the authors have adequately addressed the points raised by that reviewer, as detailed below.

- 1) The Berendsen thermostat was not actually used. This was a mistake in the description of the simulation procedures, which is now corrected.
- 2) Computation of error bars. The authors use an approximate way of computing error bars, which they call an estimate of the error. The main purpose of computing error bars in any measurement is to provide a measure of significance of the mean value and differences between various mean values. (Unless the work is on the statistics themselves (and distributions about the mean), but this is not the case here). Therefore, although not exact, I think the authors' approach serves the main purpose.
- 3) High-temperature simulations. Water evaporation actually does not occur in these simulations, as the authors explain. A more serious concern would be effect on peptide unfolding on the membrane, but that is precluded by the very large Gibbs energy difference between the folded and unfolded states in the membrane (arising from the need to form H-bonds in a medium without water).
- 4) Choice of a PC membrane. Indeed PC does not mimic the lipid component of a bacterial membrane. Would the distributions of the peptides structures observed here be the same in a PE/PG membrane? Probably not. However, what we know from experiment is that the structures of antimicrobial peptides tend not to be vastly different in different membranes; what varies is usually the weight of different states in the structural distributions observed. The use of PC, as the authors indicate, has the advantage that the results of the MD simulations can be compared to experiments (which have mainly used PC).
- 5) Equilibrium distributions of oligomers and transmembrane voltage. I think the authors have done a thorough job in ensuring that the distributions of oligomers reflect equilibrium populations. Certainly, a transmembrane voltage could alter things, but I think the authors' justification is appropriate.
- 6) Minor issues: In my opinion, the authors adequately addressed the minor issues raised by this reviewer.

Reviewer #2 (Remarks to the Author)

The authors have responded well to the comments and suggestions of the reviewers. However, perhaps they could make a comment in the Extended Data (Fig 1d) that P15A induces similar 'but

less leakage as WT'; and Fig 2 (line 454) and in line 310 (Methods) insert 100% RH rather than 100% hydration.

Reply to REVIEWERS' COMMENTS:

Reviewer #1 (Remarks to the Author):

A. Response to my comments.

The authors have adequately responded to my questions. One minor remark related to my point #2: a molar scale was used to express concentrations in water, but the K_x as defined by the authors is actually a mole fraction partition coefficient (mole fraction of peptide in the bilayer divided by mole fraction in water), not a molar partition coefficient (which would be moles of peptide per liter of bilayer divided by moles of peptide per liter of water). The two coefficients differ by the ratio of the molar volumes of water to lipid. As it is, the Gibbs energies of transfer calculated in the paper should be directly comparable with calculations using the Wimley-White scale.

Reply: Yes, that was our aim and is indeed correct. Thank you for reviewing our manuscript.

B. Response to the comments of reviewer #3.

At the request of the editor, I have read the authors' response to reviewer #3. In my opinion, the authors have adequately addressed the points raised by that reviewer, as detailed below.

1) The Berendsen thermostat was not actually used. This was a mistake in the description of the simulation procedures, which is now corrected.

2) Computation of error bars. The authors use an approximate way of computing error bars, which they call an estimate of the error. The main purpose of computing error bars in any measurement is to provide a measure of significance of the mean value and differences between various mean values. (Unless the work is on the statistics themselves (and distributions about the mean), but this is not the case here). Therefore, although not exact, I think the authors' approach serves the main purpose.

3) High-temperature simulations. Water evaporation actually does not occur in these simulations, as the authors explain. A more serious concern would be effect on peptide unfolding on the membrane, but that is precluded by the very large Gibbs energy difference between the folded and unfolded states in the membrane (arising from the need to form H-bonds in a medium without water).

4) Choice of a PC membrane. Indeed PC does not mimic the lipid component of a bacterial membrane. Would the distributions of the peptides structures observed here be the same in a PE/PG membrane? Probably not. However, what we know from experiment is that the structures of antimicrobial peptides tend not to be vastly different in different membranes; what varies is usually the weight of different states in the structural distributions observed. The use of PC, as the authors indicate, has the advantage that the results of the MD simulations can be compared to experiments (which have mainly used PC).

Reply: We are currently performing similar simulation in anionic membranes.

5) Equilibrium distributions of oligomers and transmembrane voltage. I think the authors have done a thorough job in ensuring that the distributions of oligomers reflect equilibrium populations. Certainly, a transmembrane voltage could alter things, but I think the authors' justification is appropriate.

6) Minor issues: In my opinion, the authors adequately addressed the minor issues raised by this reviewer.

Reviewer #2 (Remarks to the Author):

The authors have responded well to the comments and suggestions of the reviewers. However, perhaps they could make a comment in the Extended Data (Fig 1d) that P15A induces similar 'but less leakage as WT'; and Fig 2 (line 454) and in line 310 (Methods) insert 100% RH rather than 100% hydration.

Reply: We have done all these changes as requested. Thank you for reviewing our manuscript.